# Self-Supervised Direct Preference Optimization for Text-to-Image Diffusion Models

**Liang Peng**[1][*][†]   **Boxi Wu**[2][*][‡]   **Haoran Cheng**[2][*]   **Yibo Zhao**[1,2]   **Xiaofei He**[1,2]

[1]FABU Inc. [2]Zhejiang University

{pengliang, wuboxi, chenghaoran}@zju.edu.cn

## Abstract

Direct preference optimization (DPO) is an effective method for aligning generative models with human preferences and has been successfully applied to fine-tune text-to-image diffusion models. Its practical adoption, however, is hindered by a labor-intensive pipeline that first produces a large set of candidate images and then requires humans to rank them pairwise. We address this bottleneck with self-supervised direct preference optimization, a new paradigm that removes the need for any pre-generated images or manual ranking. During training, we create preference pairs on the fly through self-supervised image transformations, allowing the model to learn from fresh and diverse comparisons at every iteration. This online strategy eliminates costly data collection and annotation while remaining plug-and-play for any text-to-image diffusion method. Surprisingly, the on-the-fly pairs produced by the proposed method not only match but exceed the effectiveness of conventional DPO, which we attribute to the greater diversity of preferences sampled during training. Extensive experiments with Stable Diffusion 1.5 and Stable Diffusion XL confirm that our method delivers substantial gains.

## 1   Introduction

Text-to-image diffusion models [1, 2, 3, 4, 5, 6, 7, 8] have emerged as a dominant paradigm for high-quality image generation conditioned on natural language prompts. They have demonstrated the ability to synthesize diverse and visually appealing images across a wide range of prompts and styles. However, while these models are typically pretrained on large-scale datasets, they can fail to align with human preferences, especially in applications requiring text-image alignment or subjective aesthetic quality. To bridge this gap, recent works have explored preference-based post-training strategies that adjust the model's outputs based on human feedback.

One prominent method in this direction is Direct Preference Optimization (DPO) [9, 10], which trains generative models using pairwise human preference data. DPO has shown strong results in aligning text-to-image models with user intent. Nevertheless, its practical adoption is hindered by a costly and rigid training pipeline: it first requires the offline generation of a large set of image candidates, followed by extensive human annotation in the form of pairwise ranking. This process is not only time-consuming and expensive but also inflexible—once the ranking data is collected, it cannot easily adapt to new prompts or domains. Furthermore, the fixed nature of the dataset may limit the diversity of learning signals, potentially affecting generalization.

In this work, we present Self-Supervised Direct Preference Optimization (Self-DPO), a novel framework that eliminates the reliance on pre-generated images and manual rankings. Instead of requiring

---

[*]Equal contribution

[†]Work was done at FABU Inc.

[‡]Corresponding author

39th Conference on Neural Information Processing Systems (NeurIPS 2025).

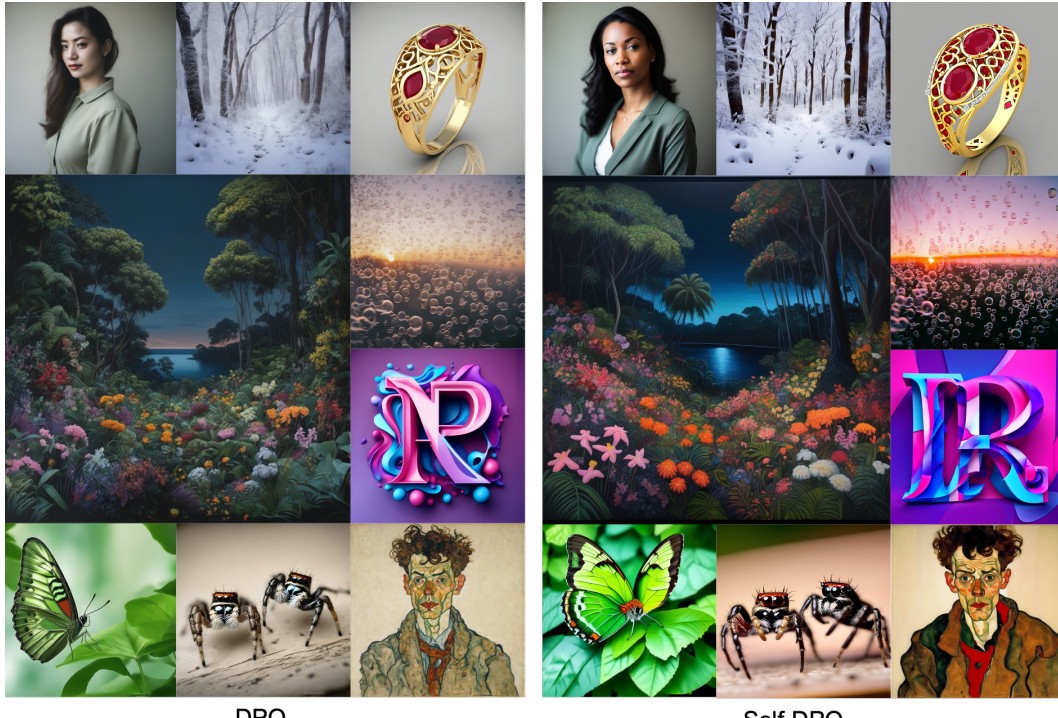

|                  DPO                  |                 Self-DPO                 |

Figure 1: We propose Self-DPO, incorporating direct preference optimization in a self-supervised manner. We provide the comparisons on SDXL base model with the same seed. Our method requires less data, yet generate more visually appealing results. Best viewed in color with zoom in.

external preference data, our method constructs training pairs dynamically using self-supervised image transformations. During each training iteration, the model first identifies a "winning image" that satisfies human-aligned quality criteria. We then generate a corresponding "losing image" by intentionally degrading the winner, either through visual-quality reductions or text–image misalignment. The resulting win–lose pair supplies an immediate preference signal, allowing the model to perform online direct preference optimization. By learning from this continuous stream of synthetically generated preference pairs, Self-DPO achieves effective human alignment.

This self-supervised, on-the-fly approach brings several key advantages. It removes the need for costly pre-generated images and human ranking efforts, enables scalable and dynamic training, and introduces greater diversity into the preference supervision signal. We summarize the data requirements in Table 1. Remarkably, we find that Self-DPO not only matches but surpasses conventional DPO in both qualitative and quantitative metrics (*e.g.*, boosting ImageReward win rate from 61 to 85). We provide some visualiza-

| Data requirements    | SFT | Self-DPO | DPO |
|----------------------|-----|----------|-----|
| Text Caption         | ✓   | ✓        | ✓   |
| Image                | ✓   | ✓        | ✓   |
| Extra Preference Image | –  | –        | ✓   |

Table 1: Post-training data requirements. Self-DPO shares the same data requirements as SFT [11]. DPO [10] requires extra preference images that are collected and annotated by humans, which is highly time-consuming and expensive.

tion results in Figure 1. Experiments on Stable Diffusion 1.5 [1] and XL [2] demonstrate consistent improvements in visual quality and text-image alignment. Our method is plug-and-play, generalizable across architectures, and easily integrable into existing diffusion model training pipelines.

## 2   Related Work

### 2.1   Text-to-Image Diffusion Models

Text-to-image diffusion models have recently attracted significant attention in generative modeling due to their ability to produce high-quality and diverse images from textual descriptions. The introduction

of Denoising Diffusion Probabilistic Models (DDPMs) [12, 13] marked a major breakthrough in this field, establishing diffusion models as a powerful generative approach. Diffusion models [12, 14, 15] operate by reversing a gradual diffusion process, where noise is incrementally added to the clean latent and then learned to be removed, ultimately synthesizing a coherent and realistic image/video. Building upon this foundation, diffusion models have become a representative paradigm in text-to-image generation, leading to models like GLIDE [16] for text-guided image editing, Imagen [17] with cascaded diffusion for high-resolution synthesis, and Stable Diffusion [1] using latent diffusion for efficient generation. To enhance image generation quality, researchers typically focus on two main directions. One approach involves architectural improvements, with the Diffusion Transformer (DiT) [3] gaining attention for its improved image fidelity, performance, and diversity. Notable advancements include PixArt-$\alpha$ [18], Hunyuan-DiT [19], and SD3 [4]. Another approach leverages supervised fine-tuning to refine text-to-image diffusion models. These methods curate datasets by integrating various strategies, such as preference models [2], pre-trained image models [20, 21, 22, 23, 24] (e.g., image captioning models), and expert-assisted data filtering [11].

## 2.2  Preference-Based Optimization Methods

In recent years, preference-based optimization has gained traction, refining models through user feedback or ranked preference pairs. In Large Language Models, Reinforcement Learning from Human Feedback (RLHF) leverages human comparisons to train a reward model that guides policy learning [25, 26]. Alternatively, direct preference optimization (DPO) fine-tunes models directly on preference data, bypassing the need for an explicit reward model while achieving comparable performance [9]. Subsequently, preference-based optimization has been applied to image generation. Some methods enhance image quality by increasing rewards for preferred outputs [27, 28, 29], while others use reinforcement learning [29, 30]. However, training reliable reward models remains challenging and computationally expensive, with over-optimization potentially leading to mode collapse, reducing diversity [31, 28]. Similarly, Direct Preference Optimization has been introduced in text-to-image generation, with Diffusion-DPO [10] demonstrating the effectiveness of optimizing on human comparison data to enhance both visual appeal and text alignment. Additionally, direct score preference optimization [32] refines diffusion models through score matching, providing a novel approach to preference learning. Several recent studies have further explored adapting preference learning techniques from large language models to fine-tune diffusion models [33, 34**?**, 35], highlighting the growing interest in aligning generative models with human preferences.

## 2.3  Self-Supervised Learning

Self-supervised learning has emerged as a pivotal paradigm in machine learning. Among its most widely used approaches are contrastive self-supervised learning and generative self-supervised learning. Contrastive self-supervised learning distinguishes representations by pulling similar instances closer while pushing dissimilar ones apart. MoCo [36] enhances training efficiency through a momentum encoder, while SimCLR [37] simplifies the process with strong data augmentations. CLIP [38] extends contrastive learning to vision-language tasks, aligning images and text in a shared latent space, enabling zero-shot transfer. Generative approaches inherently follow unsupervised or self-supervised learning principles, training without labeled data to model the underlying distributions of the input. GANs [39] utilize adversarial training to generate realistic data, while VAEs [40] encode data into a structured latent space for controlled synthesis. VQ-VAE [41] introduces discrete latent representations for high-quality generation. MAE [42] leverages masked image modeling to learn rich visual features. More recently, denoising diffusion models [13] have demonstrated impressive results by iteratively adding and removing noise, learning robust representations in a self-supervised manner. Self-supervised learning enables training without manually labeled data, laying the foundation for future advancements in representation learning, generative modeling, and multimodal understanding.

## 3  Methods

### 3.1  Preliminary

**Diffusion Models.** Denoising Diffusion Probabilistic Models [13] represent the image generation process as a Markovian process. Let $\boldsymbol{x}_0 \in \mathbb{R}^d$ be the data point and $q(\boldsymbol{x}_t|\boldsymbol{x}_{t-1})$ denote the forward

process, where noise is added to the data at each timestep $t$. The forward process is defined as:

$$q(\boldsymbol{x}_t|\boldsymbol{x}_{t-1}) = \mathcal{N}(\boldsymbol{x}_t; \sqrt{1-\beta_t}\boldsymbol{x}_{t-1}, \beta_t\mathbb{I}), \tag{1}$$

where $\beta_t$ is a schedule that controls the variance of noise added at each timestep, and $\mathcal{N}(\cdot)$ denotes the normal distribution. The forward process gradually adds noise, with $\boldsymbol{x}_T$ being pure noise after $T$ timesteps. The reverse process aims to learn the distribution $p_\theta(\boldsymbol{x}_{t-1}|\boldsymbol{x}_t)$, which represents the process of denoising and generating data from pure noise. The reverse process can be parameterized as:

$$p_\theta(\boldsymbol{x}_{t-1}|\boldsymbol{x}_t) = \mathcal{N}(\boldsymbol{x}_{t-1}; \mu_\theta(\boldsymbol{x}_t, t), \Sigma_\theta(\boldsymbol{x}_t, t)), \tag{2}$$

where $\mu_\theta(\boldsymbol{x}_t, t)$ and $\Sigma_\theta(\boldsymbol{x}_t, t)$ are the mean and covariance learned by the model at each timestep $t$. The model is trained by minimizing the following objective:

$$\mathcal{L}_{\text{diffusion}} = \mathbb{E}_{t, \boldsymbol{x}_0, \epsilon} \left[ \|\epsilon - \epsilon_\theta(\boldsymbol{x}_t, t)\|^2 \right], \tag{3}$$

where $\epsilon$ is the noise added at each timestep, and $\epsilon_\theta(\boldsymbol{x}_t, t)$ is the model's predicted noise. The model is trained to predict this noise accurately at each step, enabling it to reverse the diffusion process and generate high-quality data.

**Reinforcement Learning from Human Feedback.** For diffusion models, human preferences at each diffusion step are modeled using a Bradley-Terry formulation [43], where the probability of preferring a "winning" sample $\boldsymbol{x}_t^w$ over a "losing" sample $\boldsymbol{x}_t^l$ for a given prompt $\boldsymbol{c}$ is defined as:

$$p_{\text{BT}}(\boldsymbol{x}_t^w \succ \boldsymbol{x}_t^l | \boldsymbol{c}) = \sigma\big(r(\boldsymbol{c}, \boldsymbol{x}_t^w) - r(\boldsymbol{c}, \boldsymbol{x}_t^l)\big), \tag{4}$$

with $\sigma, r, \boldsymbol{c}$ representing the sigmoid function, reward model, and prompt, respectively. Subsequently, by conceptualizing the diffusion denoising process as a multi-step Markov Decision Process, the generative model is fine-tuned via reinforcement learning. The training objective [10, 44] is formulated as:

$$\mathcal{L}_{\text{rlhf}} = \mathbb{E}_{\boldsymbol{c}\sim D} \mathbb{E}_{p_\theta(\boldsymbol{x}_{0:T}|\boldsymbol{c})} \sum_{t=0}^{T-1} r(\boldsymbol{c}, \boldsymbol{x}_t) - \lambda\, \mathbb{D}_{\text{KL}}\big(p_\theta(\boldsymbol{x}_{0:T}|\boldsymbol{c}) \,\|\, p_{\text{ref}}(\boldsymbol{x}_{0:T}|\boldsymbol{c})\big), \tag{5}$$

where $p_{\text{ref}}(\boldsymbol{x}_{0:T}|\boldsymbol{c})$ is the distribution from the pretrained diffusion model and $\lambda$ controls the influence of the KL divergence regularization term.

**Direct Preference Optimization (DPO).** DPO streamlines RLHF by using the learning policy's log likelihood to imp licitly encode the reward. In text-to-image diffusion models, this leads to a step-wise reward defined as:

$$r(\boldsymbol{c}, \boldsymbol{x}_t) = \lambda \log \frac{p_\theta(\boldsymbol{x}_t \mid \boldsymbol{x}_{t+1}, \boldsymbol{c})}{p_{\text{ref}}(\boldsymbol{x}_t \mid \boldsymbol{x}_{t+1}, \boldsymbol{c})}. \tag{6}$$

Recent works [10, 32] follow this line and adapt it to diffusion models. They optimize the model $p_\theta$ based on the Bradley-Terry model [43], leading to an objective function:

$$\mathcal{L}_{\text{Diffusion-DPO}} = -\mathbb{E}\Big[\log \sigma\Big(\lambda \log \frac{p_\theta(\boldsymbol{x}_t^w \mid \boldsymbol{x}_{t+1}^w, \boldsymbol{c})}{p_{\text{ref}}(\boldsymbol{x}_t^w \mid \boldsymbol{x}_{t+1}^w, \boldsymbol{c})} - \lambda \log \frac{p_\theta(\boldsymbol{x}_t^l \mid \boldsymbol{x}_{t+1}^l, \boldsymbol{c})}{p_{\text{ref}}(\boldsymbol{x}_t^l \mid \boldsymbol{x}_{t+1}^l, \boldsymbol{c})}\Big)\Big], \tag{7}$$

where the winning and losing samples $(\boldsymbol{x}_t^w, \boldsymbol{x}_t^l)$ and the prompt $\boldsymbol{c}$ are drawn from the dataset, and timestep $t$ is uniformly sampled from the diffusion process. This formulation effectively aligns the learning policy with human preference signals embedded in the reference model.

### 3.2 Self-DPO for Text-to-image Diffusion Models

Inspired by DPO's effective alignment with human preferences at the image level [10], our work aims to extend this formulation into the standard post-training process for diffusion models (*e.g.*, SFT). Unfortunately, a direct application of DPO is not feasible because it requires collecting image pairs (typically generated by different models with the same prompt or by using different seeds with the same model—along with their associated manual rankings). This approach does not align with the conventional fine-tuning pipeline and incurs significant additional costs. To overcome this limitation, we propose Self-DPO, which generates preference image pairs in a self-supervised manner.

In each training iteration, we denote the text-image pair as $(\boldsymbol{c}, \boldsymbol{x})$, where the image component is regarded as the wining image $\boldsymbol{x}^w$. Traditional DPO methods [10, 9] require generating image pairs corresponding to the same prompt, followed by manual selection of the preferred (wining) and less preferred (losing) images, denoted as $\boldsymbol{x}^w$ and $\boldsymbol{x}^l$, respectively. Our method removes such cumbersome

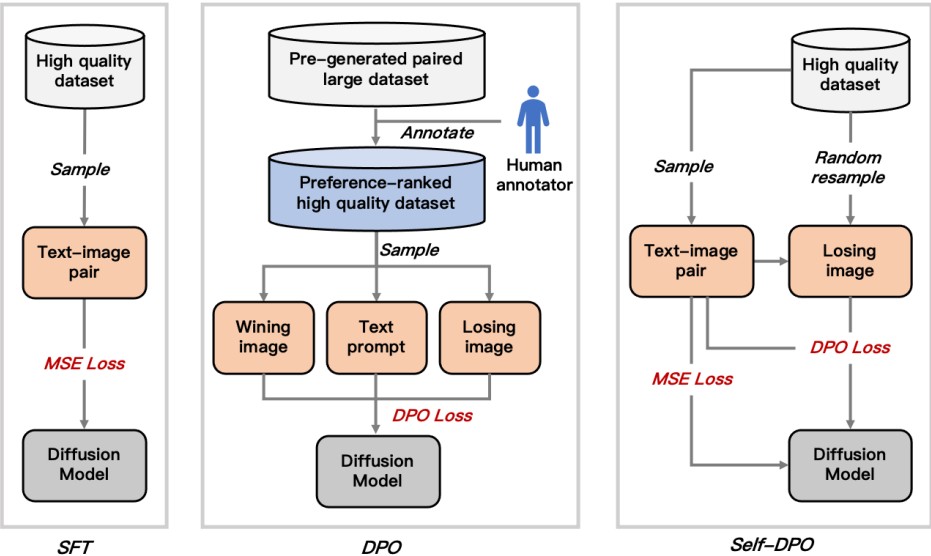

Figure 2: Different post-training processes. We generate the "losing" images self-supervisedly, enabling direct preference optimization without extra collecting and ranking steps. This lightweight procedure eliminates the substantial overhead of conventional DPO while retaining the same data requirements as standard SFT. Best viewed in color with zoom in.

steps. Because every image in the curated high-quality dataset already satisfies human preference, each can be regarded as a winner $\boldsymbol{x}^w$. We obtain a corresponding self-supervised losing sample by deliberately degrading the winner: $\boldsymbol{x}^{sl} = \textbf{Downgrade}(\boldsymbol{x}^w)$. Inspired by [10], the self-supervised direct preference loss is:

$$\mathcal{L}_{\text{Self-DPO}} = -\log \sigma \left( C \left( \left( \|\boldsymbol{\epsilon}_\theta(\boldsymbol{x}_t^w, t) - \boldsymbol{\epsilon}^w\|_2^2 - \|\boldsymbol{\epsilon}_\theta(\boldsymbol{x}_t^{sl}, t) - \boldsymbol{\epsilon}^{sl}\|_2^2 \right) \right. \right.$$
$$\left. \left. - \left( \|\boldsymbol{\epsilon}_{\text{ref}}(\boldsymbol{x}_t^w, t) - \boldsymbol{\epsilon}^w\|_2^2 - \|\boldsymbol{\epsilon}_{\text{ref}}(\boldsymbol{x}_t^{sl}, t) - \boldsymbol{\epsilon}^{sl}\|_2^2 \right) \right) \right),$$
$$where \quad \boldsymbol{x}^{sl} = \textbf{Downgrade}(\boldsymbol{x}^w) \quad (8)$$

where $C$ and $\boldsymbol{\epsilon}^{sl}$ refer to a scale factor and the noise corresponding to losing images, and $\boldsymbol{\epsilon}_{\text{ref}}$ is the reference model. In our experiments, the **Downgrade** operation can be simply performed by randomly selecting images from the training dataset. For each self-generated image pair, the winning sample closely aligns with the prompt, whereas the losing sample fails to correspond to the description. Surprisingly, this simple manner brings significant improvements to the model. We also compare different degradation strategies in the experiments. The training process overview is shown in Figure 2. The final loss is as follows:

$$\mathcal{L} = \lambda_1 \mathcal{L}_{\text{MSE}} + \lambda_2 \mathcal{L}_{\text{Self-DPO}} \quad (9)$$

We empirically set $\lambda_1$ and $\lambda_2$ to $0.5, 0.5$, respectively.

## 4 Experiments

### 4.1 Setup

**Implement Details:** Following Diffusion-DPO [10], for the SD1.5 [1] experiments, AdamW [47] is utilized, while SDXL [2] training is conducted with Adafactor [48] to conserve memory. Following the official implementation in Diffusion-DPO [10], $C$ in Equation 8 is set to $-2500$. For SD 1.5, a batch size of 2048 pairs (resolution: $512 * 512$) is maintained by training across 4 NVIDIA A100 GPU. Each GPU handles 8 pairs locally with gradient accumulation

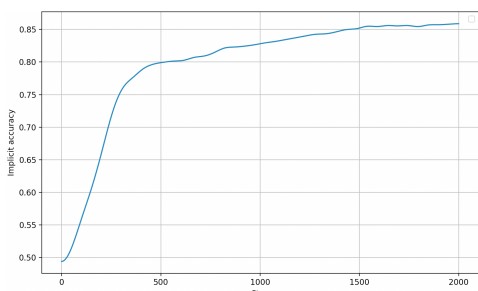

Figure 5: Implicit accuracy during the training stage.

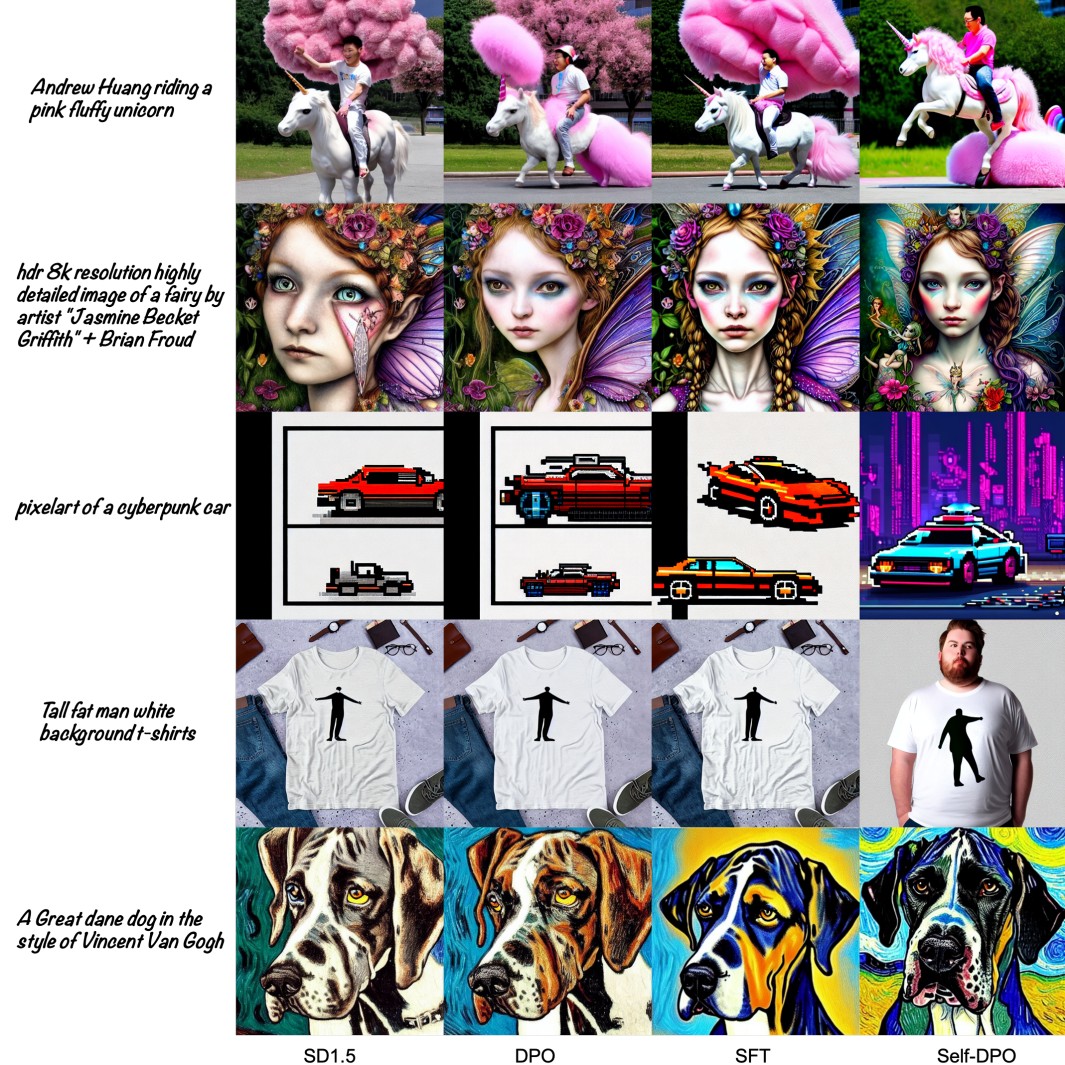

Figure 3: Qualitative comparisons with the SD1.5 base model. All results are generated with the same random seed. Comparing with SFT and DPO [10], the model trained by Self-DPO demonstrates superior text prompt alignment. It also shows more appealing visual quality, especially in layout, colors, and details. Best viewed in color.

over 64 steps. For SDXL, considering the resource limitation, we use the total batch size of 96 pairs (resolution: $1024 * 1024$). Training is performed at fixed square resolutions. We use a learning rate of 1e-6 coupled with a 25% linear warmup.

**Training Dataset:** Our training data is sourced from the Pick-a-Pic V2 dataset[45], which keeps the same to Diffusion-DPO [10]. It contains pairwise preference annotations for images generated by Dreamlike (a fine-tuned variant of SD1.5), SD2.1, and SDXL. These prompts and preferences were collected from users of the Pick-a-Pic web application. ***Please note that Self-DPO only uses the preference image and associated text in the dataset, instead of using the whole manually annotated image pairs.***

**Evaluation:** We conduct evaluation on three datasets: Pick-a-Pic V2 [45] validation set (contains 500 prompts), PartiPrompts [46] (contains 1632 prompts, including diverse categories and challenge aspects), and HPDv2 [24] (contains 3200 prompts, including anime, concept art, paintings and photo). We compare Self-DPO with three different type baselines, *i.e.*, base models (Stable Diffusion 1.5 (SD1.5) and Stable Diffusion XL (SDXL)), SFT models, DPO models, where DPO models are the

Figure 4: Qualitative comparisons with the SDXL base model. All results are generated using the same random seed. Please note that the training dataset (Pick-a-Pic V2 [45]) used for fine-tuning is obtained from a SD1.5 variant, SD2.1, and SDXL models. Consequently, directly fine-tuning (SFT) on this dataset does not lead to improvements—and may even result in degraded performance (*e.g.*, as shown in the second row). In contrast, DPO [10] optimizes the model by leveraging preference relationships with pre-ranked pairs, thereby avoiding this issue. Self-DPO uses the same data requirements as SFT but yields significant improvements in both text prompt alignment and visual quality, demonstraing its effectiveness. Best viewed in color.

publicly released from Diffusion-DPO [10]. For evaluation metrics, we employ the popular PickScore [45], Aesthetics [49], CLIP [38], HPS V2 [24], and ImageAward [50] scores. For convenience of comparison, we scale the scores to fit a similar range (×*100 for PickScore, CLIP, HPS, and ImageAward, ×10 for Aesthetics*). **PickScore** is a caption-sensitive scoring model, originally trained on Pick-a-Pic (v1), that estimates the perceived image quality by humans. **Aesthetics** assesses the visual appeal of an image, considering factors such as lighting, color harmony, composition, and overall artistic quality. **CLIP** measures the semantic alignment between an image and a corresponding text prompt. By computing the cosine similarity between the image and text embeddings, this score evaluates how well the image content matches the provided textual description. **Human Preference Score (HPS V2)** is a metric designed to align with human judgments of image quality, particularly in the context of text-to-image synthesis. **ImageAward** quantifies the quality, aesthetic appeal, or alignment of a generated image with respect to desired attributes. It is typically derived from a reward model trained on human preference data.

Table 2:

| Datasets | Methods | | SD1.5 | | | | | SDXL | | | | |
|---|---|---|---|---|---|---|---|---|---|---|---|---|
| | | | P.S. | Aes. | CLIP | HPS | I.R. | P.S. | Aes. | CLIP | HPS | I.R. |
| Pick-a-Pic V2 | Base | Avg score | 20.57 | 53.15 | 32.58 | 26.17 | -14.81 | 22.10 | **60.01** | 35.86 | 26.83 | 50.62 |
| | SFT | | 21.10 | **56.35** | 33.75 | 27.03 | 45.03 | 21.48 | 57.84 | 35.67 | 26.67 | 30.89 |
| | DPO | | 20.91 | 54.07 | 33.19 | 26.46 | 4.13 | **22.57** | 59.93 | 37.30 | 27.30 | 81.14 |
| | Self-DPO | | **21.23** | **56.35** | **34.79** | **27.33** | **71.00** | 22.34 | 59.97 | **37.53** | **27.89** | **103.96** |
| | SFT | Win rate | 75.00 | 77.20 | 60.40 | 90.20 | 80.00 | 19.40 | 31.80 | 47.00 | 44.60 | 42.4 |
| | DPO | | 73.80 | 60.00 | 60.00 | 71.80 | 61.00 | **72.60** | 47.20 | **63.00** | 79.80 | 69.8 |
| | Self-DPO | | **78.60** | **77.80** | **68.40** | **94.20** | **85.20** | 60.80 | **50.80** | 62.40 | **93.80** | **79.2** |
| PartiPrompts | Base | Avg score | 21.39 | 53.13 | 33.21 | 26.79 | 1.48 | 22.63 | 57.69 | 35.77 | 27.33 | 69.78 |
| | SFT | | 21.75 | **55.31** | 33.93 | 27.57 | 50.75 | 22.02 | 56.41 | 35.31 | 27.13 | 47.29 |
| | DPO | | 21.61 | 53.58 | 33.88 | 26.98 | 21.43 | **22.90** | 57.85 | 36.95 | 27.73 | 103.36 |
| | Self-DPO | | **21.84** | 55.09 | **35.11** | **27.84** | **75.66** | 22.79 | **58.69** | **37.00** | **28.30** | **117.50** |
| | SFT | Win rate | 67.28 | **70.89** | 53.43 | 85.42 | 73.35 | 21.38 | 38.11 | 45.10 | 43.75 | 40.93 |
| | DPO | | 67.10 | 57.17 | 56.74 | 61.83 | 63.05 | **63.42** | 53.62 | **62.32** | 73.10 | 68.44 |
| | Self-DPO | | **69.85** | 68.50 | **63.24** | **89.40** | **81.00** | 56.19 | **60.48** | 60.17 | **92.16** | **76.84** |
| HPD V2 | Base | Avg score | 20.84 | 54.32 | 33.96 | 26.84 | -11.79 | 22.78 | 61.34 | 37.68 | 27.68 | 78.27 |
| | SFT | | 21.57 | **57.41** | 35.26 | 27.89 | 57.74 | 22.24 | 60.08 | 37.39 | 27.76 | 66.62 |
| | DPO | | 21.30 | 55.80 | 34.68 | 27.22 | 13.24 | **23.18** | **61.35** | **38.45** | 28.14 | 102.74 |
| | Self-DPO | | **21.58** | 57.10 | **36.30** | **28.11** | **76.13** | 22.98 | 61.30 | 38.35 | **28.77** | **110.67** |
| | SFT | Win rate | **79.53** | **75.31** | 59.34 | 90.10 | 81.16 | 23.47 | 37.28 | 46.63 | 58.22 | 45.81 |
| | DPO | | 75.72 | 66.28 | 57.56 | 72.43 | 64.69 | **72.66** | **50.28** | **58.69** | 80.56 | 69.78 |
| | Self-DPO | | **79.53** | 74.03 | **68.47** | **92.49** | **85.19** | 58.78 | 48.06 | 55.65 | **94.97** | **72.50** |

Table 2: Quantitative comparisons. We compare different fine-tuning methods (SFT, DPO [10], and Self-DPO) on SD 1.5 and SDXL base models over three datasets (Pick-a-Pic V2 [45], PartiPrompts [46], and HPDv2 [24]). "P.S." refers to PickScore, "Aes." is Aesthetics, and "I.R." denotes ImageAward. For the SD1.5 base model, our method achieves the best performance across most metrics. In contrast, for the SDXL base model, we observe that SFT clearly degrades performance. This is likely due to the fact that the training dataset (Pick-a-pic V2 [45]) used for fine-tuning is derived from SD1.5 variant, SD2.1, and SDXL models. Interestingly, our method still achieves competitive results compared to DPO, which utilizes the full dataset and optimizes the model by leveraging preference relationships with pre-ranked pairs. These results underscore the effectiveness and robustness of our approach.

## 4.2 Quantitative Results

We provide quantitative comparisons in Table 2. We compare our method with SFT and DPO [10]. For the SD1.5 base model, our method significantly outperforms the alternatives. For example, Self-DPO achieves a CLIP score of 34.79 on the Pick-a-Pic V2 dataset—an improvement of +2.21 over the base model—whereas SFT and DPO yield improvements of +1.17 and +0.61, respectively. In terms of overall human preference metrics, Self-DPO delivers substantially higher gains, improving the base model from –14.81 to 71.00 on the ImageReward metric, which far exceeds the improvements observed with SFT (45.03) and DPO (4.13).

Notably, the win rate of Self-DPO reaches 94.20 on the HPS metric. Results on other datasets (PartiPrompts and HPD v2) further confirm these improvements. In contrast, results on the SDXL base model show a slightly different scenario, particularly with SFT. We observe that SFT substantially degrades the performance of the base model. This degradation is likely due to the

| Methods | P.S. | Aes. | CLIP | HPS | I.R. |
|---|---|---|---|---|---|
| Base model | 20.57 | 53.15 | 32.58 | 26.17 | -14.81 |
| DPO | 20.91 | 54.07 | 33.19 | 26.46 | 4.13 |
| w/ Blur | 20.80 | 55.16 | 33.22 | 26.57 | 4.26 |
| w/ Random grid | 20.85 | 55.86 | 32.71 | 26.64 | 24.21 |
| w/ Random image | **21.23** | **56.35** | **34.79** | **27.33** | **71.00** |

Table 3: Ablation on different downgrade manners. The "Random image" row achieves most significant improvements, implying that degrading the image quality can be detrimental.

fact that, while the training dataset (comprising generations from an SD1.5 variant, SD2.1, and SDXL) is of much higher quality than that used for SD1.5, it does not exhibit clear superiority for SDXL. Nevertheless, the model trained by Self-DPO still demonstrates significantly better performance compared to both SFT and the base model. On the PickScore, Aesthetics, and CLIP metrics, Self-DPO achieves results comparable to DPO, and it attains superior scores on the HPS and ImageRe-

ward metrics. These findings validate our hypothesis that image-level learning benefits text-to-image diffusion models and highlight the superiority of Self-DPO. We also employ UnifiedReward [51] as the VLM evaluation metric, a state-of-the-art reward model designed for multimodal understanding and generation tasks. Built upon strong VLM models [52, 53, 54, 55], it supports pointwise scoring to align model outputs with human preferences. The results are shown in Table 4.4. We further validate our method's effectiveness by using a new real-world T2I evaluation setting. Specifically, we conduct experiments on a subset of the LAION-COCO dataset containing 400k text-image pairs. Unlike carefully curated datasets, we do not manually design the data distribution or employ re-captioning techniques to refine prompts. The results are shown in Table 4.4. While naive SFT degrades performance, our Self-DPO method achieves consistent improvements. Please note that we cannot provide a DPO baseline as it requires additional paired images and annotations for the same prompts, which are not available in regular text-image dataset.

## 4.3 Qualitative Results

We provide qualitative comparisons in Figure 3 and Figure 4 for SD1.5 and SDXL, respectively. All quantitative and qualitative results are generated using the same random seed. We observe that Self-DPO consistently improves over other models, particularly in terms of producing more vivid colors and better adherence to text prompts. For example, in the third row of Figure 3, only Self-DPO successfully reveals the concept "cyberpunk". In the fourth row of Figure 4, both DPO and Self-DPO capture the intended meaning of the prompt, but Self-DPO exhibits a more appealing visual quality. These quantitative and qualitative results confirm the effectiveness of our approach. We show the implicit win rate to measure the optimization process in Figure 5. Specifically, the implicit win rate during training refers to the probability that the model prefers the winning image over the losing image. In other words, a higher implicit win rate indicates a stronger preference for the winning image. As training progresses, it steadily increases and eventually reaches around 0.85. This demonstrates that the model gradually learns to distinguish between the wsinning and losing images, and prefers to generate the winning image.

## 4.4 Ablation

We conduct ablation studies on various downgrade approaches, as shown in Table 3. In particular, we examine two additional methods—namely, blur (which involves downsampling and upsampling the image by a factor of 4) and random grid (which divides the image into an $8 \times 8$ grid and randomly swaps two grids).

| Methods | P.S. | Aes. | CLIP | HPS | I.R. |
|---|---|---|---|---|---|
| Base model | 20.57 | 53.15 | 32.58 | 26.17 | -14.81 |
| DPO | 20.91 | 54.07 | 33.19 | 26.46 | 4.13 |
| Self-DPO w/o MSE | **21.26** | 56.22 | **35.00** | 27.31 | 67.78 |
| Self-DPO | 21.23 | **56.35** | 34.79 | **27.33** | **71.00** |

Table 4: Ablation on MSE loss. When removing MSE loss, Self-DPO still shows comparable performance and demonstrates the superiority to DPO.

We observe that these two downgrade methods lead to worse performance, indicating that significantly degraded image quality can be harmful. Additionally, we remove the MSE loss and report the results in Table 4. Interestingly, the performance does not drop noticeably and consistently performs better than DPO, further confirming the effectiveness of Self-DPO.

| Methods | P.S. | Aes. | CLIP | HPS | I.R. |
|---|---|---|---|---|---|
| Base | 20.57 | 53.15 | 32.58 | 26.17 | -14.81 |
| SFT | 20.49 | 52.63 | 31.83 | 26.13 | -23.95 |
| Self-DPO | 20.68 | 52.87 | 34.31 | 26.52 | 17.11 |

Table 5: Results on a real-world image-text dataset.

| Base model | Base | SFT | DPO | Self-DPO |
|---|---|---|---|---|
| SD1.5 | 2.44 | 2.62 | 2.53 | 2.72 |
| SDXL | 2.96 | 2.73 | 3.10 | 3.07 |

Table 6: Comparisons on UnifiedReward.

## 5 Limitation and Future Work

In the present implementation, we construct losing samples by randomly sampling images from the training set. This manner can become unreliable when the dataset is small, as the resulting

losers may fail to provide sufficiently informative preference signals. Future work therefore can explore more sophisticated self-supervised strategies for synthesizing losing images, such as more content-aware perturbations or adversarial degradations that better challenge the model. Furthermore, our approach is built on direct preference optimization, which is an offline reinforcement-learning paradigm. Extending the self-supervised pairing concept to online policy-gradient or actor-critic frameworks represents another promising direction.

# 6 Conclusion

In this paper, we propose Self-Supervised Direct Preference Optimization (Self-DPO), a fully self-supervised framework that aligns generative models with human preferences without requiring pre-generated image pools or manual rankings. By synthesizing win–lose pairs on the fly, Self-DPO not only removes costly data-collection steps but also exposes the model to a broader and more diverse set of preference signals. At every training iteration, it constructs its own preference pairs through controlled degradations of high-quality images and immediately updates the model via preference learning. Extensive experiments across multiple datasets and base architectures demonstrate that Self-DPO consistently delivers superior performance, validating its effectiveness and versatility.

## Acknowledgments

This research was supported by The National Nature Science Foundation of China (Grant Nos: 62402417, 62273302,62036009, 61936006), in part by the Key R&D Program of Ningbo (Grant Nos: 2024Z115, 2025Z035), in part by Yongjiang Talent Introduction Programme (Grant No: 2023A-197-G).

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
