# OpenReview forum: "Self-Supervised Direct Preference Optimization for Text-to-Image Diffusion Models"
_NeurIPS.cc/2025/Conference — NeurIPS 2025 poster_

### Official Review · Reviewer_4ghY · 2025-06-01

**Clarity:** 3
**Significance:** 3
**Originality:** 3
**Rating:** 5
**Confidence:** 4

**Summary:**

The paper introduces Self-Supervised Direct Preference Optimization (Self-DPO), a training paradigm for aligning text-to-image diffusion models with human preferences—without requiring any manually ranked data or pre-generated images dataset. Instead, the method dynamically constructs training pairs by identifying a "good" image, and a corresponding "bad" image by downgrading the winner. In practice, the authors choose to downgrade the winning image by selecting a random image instead, which lacks text faithfulness. Importantly, this approach surpasses the more expensive DPO baseline on several datasets, metrics, and base models. Additionally, the authors experiment with different downgrading operations such as blur, random image, and patch mixing.

**Questions:**

1. The authors make use of the "preference image" in the Pick-a-Pic dataset. However, the claim is that the algorithm does not need annotation. To better support this claim, it can be helpful for the authors to sample between the preferred image and the rejected image. Will the approach work using random images from pick-a-pic?
2. Will the algorithm work with real images rather than generated images?
3. Will the approach work for SOTA models such as FLUX? If it is unclear how to perform such an experiment, it can be useful to add this to the limitations section.
4. For different downgrading operations, are you able to analyze and identify different improvement axis? this can also be answered qualitatively.

**Ethical Concerns:**

["NO or VERY MINOR ethics concerns only"]

**Final Justification:**

Solid paper, see the other sections as for why I gave this score.

**Limitations:**

yes

**Paper Formatting Concerns:**

ok

**Quality:**

3

**Strengths And Weaknesses:**

Strengths:
1. The core idea of selecting a winning image and utilizing a simple and efficient heuristic to get a losing image, and then applying DPO is simple and very elegant.
2. The effectiveness of this approach when compared to DPO, combined with the limited resources required to apply it, make it both appealing and accessible.
3. The paper is easy to follow.
4. The experimental setup is extensive and contains multiple models, datasets, metrics, etc.

Weaknesses:
1. The authors make use of the preference image in the Pick-a-Pic dataset. However, the claim is that the algorithm does not need annotation. To support the claim the authors should use random images from the Pick-a-Pic dataset, rather than the preference image.

---

> ### Author Rebuttal · Authors · 2025-07-30
>
> **W1 & Q1:The authors make use of the preference image in the Pick-a-Pic dataset. However, the claim is that the algorithm does not need annotation. To support the claim the authors should use random images from the Pick-a-Pic dataset, rather than the preference image; Will the approach work using random images from pick-a-pic?**
>
> Yes, the method performs effectively when using randomly selected images from the pick-a-pic dataset. As shown follows:
>
> |   Models    | PickScore | Aesthetics | CLIP | HPS | ImageReward
> |-----------------------|-----------|-----------|-----------|-----------|-----------|
> |  SD1.5 | 20.57 |53.15|32.58 |26.17| -14.81
> |  SD1.5 + DPO   |20.91 |54.07 |33.19 |26.46 | 4.13
> |  SD1.5 + Self-DPO  |21.23 |56.35 |34.79 |27.33 |71.00
> |  SD1.5 + Self-DPO (Random images) |21.20 | 56.25 |34.49 |27.39 | 59.18
>
> The results demonstrate that the random selection approach achieves comparable performance to the standard Self-DPO method across most metrics. We appreciate your feedback and will incorporate these findings into the revision.
>
> **Q2:Will the algorithm work with real images rather than generated images?**
>
> Yes, the method also works with real images. Specifically, we conduct experiments on a subset of the LAION-COCO dataset containing ~400k text-image pairs. Unlike carefully curated datasets, we do not manually design the data distribution or employ re-captioning techniques to refine prompts.
>
> |   Models    | PickScore | Aesthetics | CLIP | HPS | ImageReward
> |-----------------------|-----------|-----------|-----------|-----------|-----------|
> |  SD1.5 | 20.57 |53.15|32.58 |26.17| -14.81
> |  SD1.5 + SFT   |20.49| 52.63 |31.83 |26.13| -23.95
> |  SD1.5 + Self-DPO  |20.68| 52.87 |34.31 |26.52 | 17.11
>
> While naive SFT degrades performance, our Self-DPO method achieves consistent improvements over the baseline across four out of five metrics.
> Please note that we cannot provide a DPO baseline as it requires additional paired images and annotations for the same prompts, which are not available in regular text-image dataset.
>
> **Q3:Will the approach work for SOTA models such as FLUX? If it is unclear how to perform such an experiment, it can be useful to add this to the limitations section.**
>
> Yes, Self-DPO works for SOTA models such as FLUX beacuase it is model-agnosic. Ideally it can be employed in most diffusion models. As an initial attempt, we performs this experiment on FLUX using the Pick-a-Pic V2 synthetic dataset. Notably, this dataset comprises image generations from SD2.1 and SDXL, which have lower quality compared to FLUX outputs. Despite this, our experiments demonstrate that the method remains effective when applied to modern architectures.
>
> |                 | PickScore | Aesthetics | CLIP | HPS | ImageReward
> |-----------------------|-----------|-----------|-----------|-----------|-----------|
> | Flux    | 22.45     | 62.25 | 32.95 | 28.81 | 102.00
> | Flux + DPO  | 22.46      |  **62.38** | 32.88 | 28.86 | 101.41
> | Flux + Self-DPO  | **22.52**     |  61.98 | **33.56** | **28.99** | **108.31**
>
> **Q4:For different downgrading operations, are you able to analyze and identify different improvement axis? this can also be answered qualitatively.**
>
> Thank you for your feedback. Each image downgrading operation involves distinct optimization dimensions. For random grid downgradtion, it varies in grid configurations and selection numbers. In such scenarios, a enhancement could involve using instance masks to relocate objects within the image, generating more feasible results with minimal fidelity loss. For blur downgradtion, it can be changed by different blur kernels and area. We appreciate your suggestion and will incorporate this discussion into the revision.

---

### Official Review · Reviewer_cdb2 · 2025-06-20

**Clarity:** 3
**Significance:** 3
**Originality:** 3
**Rating:** 5
**Confidence:** 5

**Summary:**

The paper proposes Self-DPO, a self-supervised approach to Direct Preference Optimization for text-to-image (T2I) diffusion models. Instead of relying on externally curated preference pairs, the method automatically constructs a “winning vs. losing” image pair by degrading the higher-quality sample. This eliminates the costly human-labelled preference data required by conventional DPO pipelines.

**Questions:**

1. The results show that Self-DPO is even better than the DPO with human annotation datasets. Could the authors explain this?

2. Random replacement often yields completely unrelated images, could the model over-weight low-level aesthetics at the expense of fidelity? A qualitative failure analysis would be valuable.

3. When you draw a random loser from the same minibatch, its prompt is different from the winner’s. Do you ensure different seeds each epoch to avoid memorization?

**Ethical Concerns:**

["NO or VERY MINOR ethics concerns only"]

**Final Justification:**

My concerns are well-addressed

**Limitations:**

Please see the Weaknesses.

**Quality:**

3

**Strengths And Weaknesses:**

### Strengths

1. By automatically “downgrading” high-quality images to obtain the losing side of each preference pair, the authors remove the expensive component of DPO—data collection and annotation. This design not only cuts labeling costs, but also enables the method to bootstrap essentially unlimited preference data from any captioned image corpus. As a result, the technique can be scaled to new domains or larger datasets without the bottleneck of crowdsourcing or labeling.

2. Experiments show consistent improvements in CLIP-Score, ImageReward, and related quality metrics—despite using fewer training samples than baseline DPO. The method therefore offers both sample efficiency and better performance.

3. The paper lists the degradation step and key hyperparameters in clear fashion. Readers can replicate the pipeline with minimal code changes to existing diffusion frameworks. The provided code in the supplementary material is also helpful.

### Weaknesses

Major:

1. While the paper reports final metrics, it omits learning-curve plots comparing Self-DPO with vanilla DPO. Without these curves, readers cannot assess convergence speed and optimization stability. Such information is important to judge whether the observed gains come from faster learning, better local minima, or merely longer training.

2. The model is trained on Pick-a-Pic, a synthetic preference dataset whose images were generated by diffusion models. Because Self-DPO no longer requires explicit preference pairs, it is crucial to demonstrate effectiveness on a standard text–image dataset. A comparison against conventional supervised fine-tuning (SFT) on such datasets would reveal whether the method’s advantages persist in real-world scenarios.

Minor:

1. Typo: Table 4, second row: extra “s”; Line 161: “wining image”->“winning image”; Figure 4 caption:“demonstraing” ->“demonstrating”

2. How were these hyperparameters (\lambda_1 and \lambda_2 in Equ.9) selected, and how sensitive are the results to them?

3. Blur and random-grid corruptions can hurt performance, whereas replacing the winner with a random image helps dramatically. Could you analyze why semantic mismatches outperform perceptual degradations?

Overall, this paper is clearly written and the idea is both simple and interesting, but addressing the weaknesses above—particularly learning curve and more analysis—will significantly strengthen the submission.

---

> ### Author Rebuttal · Authors · 2025-07-30
>
> **W1:While the paper reports final metrics, it omits learning-curve plots comparing Self-DPO with vanilla DPO. Without these curves, readers cannot assess convergence speed and optimization stability. Such information is important to judge whether the observed gains come from faster learning, better local minima, or merely longer training.**
>
> All baseline models are trained for the same number of iterations. We observe that the learning curves can demonstrate consistent improvement over training. Unfortunately, due to the rebuttal policy, we are unable to include visualizations in this rebuttal. The final figures and detailed descriptions will be provided in the revision. We appreciate your feedback.
>
> **W2:It is crucial to demonstrate effectiveness on a standard text–image dataset. A comparison against conventional supervised fine-tuning (SFT) on such datasets would reveal whether the method’s advantages persist in real-world scenarios.**
>
> Specifically, we conduct experiments on a subset of the LAION-COCO dataset containing ~400k text-image pairs. Unlike carefully curated datasets, we do not manually design the data distribution or employ re-captioning techniques to refine prompts.
>
> |   Models    | PickScore | Aesthetics | CLIP | HPS | ImageReward
> |-----------------------|-----------|-----------|-----------|-----------|-----------|
> |  SD1.5 | 20.57 |53.15|32.58 |26.17| -14.81
> |  SD1.5 + SFT   |20.49| 52.63 |31.83 |26.13| -23.95
> |  SD1.5 + Self-DPO  |20.68| 52.87 |34.31 |26.52 | 17.11
>
> While naive SFT degrades performance, our Self-DPO method achieves consistent improvements over the baseline across four out of five metrics.
> Please note that we cannot provide a DPO baseline as it requires additional paired images and annotations for the same prompts, which are not available in regular text-image dataset.
>
> **Minor1:Typo.**
>
> We will fix it in the revision, thank you.
>
> **Minor2: How were these hyperparameters ($\lambda_1$ and $\lambda_2$ in Equ.9) selected, and how sensitive are the results to them?**
>
> We present an initial ablation study on the impact of removing the MSE loss in the main text. Here, we provide an expanded analysis comparing different weight ratios between the Self-DPO and MSE components:
>
> |     $\lambda_2$ : $\lambda_1$         | PickScore | Aesthetics | CLIP | HPS | ImageReward
> |-----------------------|-----------|-----------|-----------|-----------|-----------|
> |  0.00 : 1.00   | 21.10 | 56.35 |33.75 |27.03 |45.03
> |  0.25 : 0.75   |21.19| 56.23 |34.71 |27.29 |66.20
> | 0.50 : 0.50  | 21.23 |56.35 |34.79 |27.33 |71.00
> | 0.75 : 0.25  | 21.24 | 56.24 |34.75 |27.34| 68.27
> | 1.00: 0.00 |21.26| 56.22 |35.00 |27.31 |67.78
>
> We have two obervations:
> First, when Self-DPO is entirely removed (0.0:1.0), the method reduces to standard SFT, yielding the lowest performance across all metrics.
> Second, introducing Self-DPO (ratios > 0.0:1.0) consistently improves results, with performance remaining stable across varying weight distributions. This suggests the Self-DPO loss plays the main role during the training process.
> These findings confirm that our method is insensitive to loss weight tuning, as the Self-DPO component primarily delievers performance gains.
>
> **Minor3: Blur and random-grid corruptions can hurt performance, whereas replacing the winner with a random image helps dramatically. Could you analyze why semantic mismatches outperform perceptual degradations?**
>
> This phenomenon can be attributed to the model's greater ease in learning blur and random-grid corruption patterns. In contrast, identifying random images proves more challenging for the model due to their similar quality, which necessitates heightened sensitivity to prompts, thereby enhancing performance. Future work could explore more diverse and reliable image degradation strategies to further optimize model capabilities.
>
> **Q1:The results show that Self-DPO is even better than the DPO with human annotation datasets. Could the authors explain this?**
>
> This can be attributed to the fact that conventional DPO relies on pre-generated offline data with limited diversity, whereas our method dynamically generates preference pairs in an online fashion. Online generation introduces greater data diversity by continuously sampling from the model's current policy distribution, addressing the static nature of offline datasets.
>
> **Q2:Random replacement often yields completely unrelated images, could the model over-weight low-level aesthetics at the expense of fidelity? A qualitative failure analysis would be valuable.**
>
> The model does not sacrifice fidelity during preference optimization. A detailed qualitative failure analysis will be included in the revision. Thanks for your advice.
>
> **Q3:When you draw a random loser from the same minibatch, its prompt is different from the winner’s. Do you ensure different seeds each epoch to avoid memorization?**
>
> Yes, each epoch generates different image pairs within the same minibatch.

---

> > ### Comment · Reviewer_cdb2 · 2025-08-03
> >
> > Thank you for the detailed rebuttal. I appreciate the clarification, but I have additional questions for further investigation.
> >
> > 1. What metrics are used to assess the optimization process in the learning curves? For performance learning, does the metric focus on task-specific accuracy?
> >
> > 2. Regarding the real-world text-image dataset construction, could the authors elaborate on the methodology used to build the dataset? Does the observed degradation of SFT stem from inherent dataset biases (e.g., distributional shifts) or from limitations in the SFT training objective?

---

> > > ### Author Response · Authors · 2025-08-03
> > >
> > > Thank you for your feedback! Here, we provide further clarification.
> > >
> > > **For question 1:**
> > > We employ both the loss curve and the implicit win rate to measure the optimization process, with the latter being closely related to preference learning. Specifically, the implicit win rate during training refers to the probability that the model prefers the winning image over the losing image. In other words, a higher implicit win rate indicates a stronger preference for the winning image. At the beginning of training, the implicit win rate is close to 0.5. As training progresses, it steadily increases and eventually reaches around 0.85. This demonstrates that the model gradually learns to distinguish between the winning and losing images, and prefers to generate the winning image.
> > >
> > > **For question 2:**
> > > We use a subset of the LAION-COCO dataset. The LAION-COCO dataset contains MS COCO-style captions for images, with a total of 600 million images from the English subset of LAION-5B. Due to its large scale, we apply simple filtering rules as in LAION-COCO-AESTHETIC, such as image size >384x384, aesthetic score >4.75, and watermark probability <0.5, resulting in approximately 8 million samples. From these, we randomly select 400,000 samples for the real-world text-to-image experiment.
> > > Regarding the degradation of SFT, since we do not carefully design the dataset distribution and the captions follow the COCO style, the performance is limited by distributional shifts. A comprehensive, well-designed, high-quality dataset is essential for SFT to perform well; therefore, in our roughly filtered dataset, SFT shows degraded performance.
> > >
> > > In contrast, Self-DPO is more robust to distributional shifts because it benefits from pairwise preference learning. Even when the diversity and quality of the dataset are not sufficient, Self-DPO can still leverage dynamically generated preference image pairs for learning. However, we also observe that Self-DPO achieves relatively lower performance compared to models trained on the Pick-a-Pic V2 dataset, demonstrating that Self-DPO can also benefit from a well-designed dataset.

---

> > > > ### Comment · Reviewer_cdb2 · 2025-08-03
> > > >
> > > > Thank you for your answers. My concerns are well-addressed; I will give a higher score in the final decision.

---

> > ### Comment · Reviewer_cdb2 · 2025-08-03
> >
> > Additionally, while SFT fails to perform well, the Self-DPO method still achieves improvements. In this setting, what enables Self-DPO to overcome the limitations of SFT?

---

### Official Review · Reviewer_vbGM · 2025-07-01

**Clarity:** 3
**Significance:** 2
**Originality:** 2
**Rating:** 4
**Confidence:** 3

**Summary:**

The paper proposes a surprisingly simple method to improve upon Diffusion-DPO. Traditionally, DPO requires a preference-pairs dataset, but the proposed method only requires a high-quality dataset. In this sense, the technique can achieve performance gains for free beyond standard SFT.

**Questions:**

1. The paper lacks a fair comparison with Diffusion-KTO [1], which is an important baseline as it also does not require paired data.

2. The paper does not clearly specify the fine-tuning method used. While using MSE loss is a natural choice, it is conceptually similar to LoRA fine-tuning, where the goal is to keep the tuned model close to the original. More ablation studies are needed to demonstrate the effectiveness of the MSE loss in this context.

3. This paper's method is simple and clean, which is good. The effectiveness of the proposed method needs more evidence.



[1] Aligning Diffusion Models by Optimizing Human Utility. NeurIPS 2024.

**Ethical Concerns:**

["NO or VERY MINOR ethics concerns only"]

**Final Justification:**

I thank the authors for the detailed feedback in the rebuttal. I thus raise my rating to 4.

**Limitations:**

yes

**Paper Formatting Concerns:**

Rows 168 and 169 are too close.

**Quality:**

3

**Strengths And Weaknesses:**

**Strengths:**

1. The paper is clear and easy to follow.
2. It provides fair comparisons with DPO across multiple benchmarks.

**Weaknesses:**

1. The paper lacks a fair comparison with Diffusion-KTO [1], which is an important baseline as it also does not require paired data.
2. More ablation studies on fine-tuning methods used and its interaction effect with MSE losses are needed to clarify the effectiveness of each component.


[1] Aligning Diffusion Models by Optimizing Human Utility. NeurIPS 2024.

---

> ### Author Rebuttal · Authors · 2025-07-30
>
> **W1 & Q1:The paper lacks a fair comparison with Diffusion-KTO, which is an important baseline as it also does not require paired data; The paper lacks a fair comparison with Diffusion-KTO, which is an important baseline as it also does not require paired data.**
>
> We apologize for the omission and provide qualitative comparisons below. The experiments were conducted using the official released model for fair comparison:
>
> |     Models       | PickScore | Aesthetics | CLIP | HPS | ImageReward
> |-----------------------|-----------|-----------|-----------|-----------|-----------|
> |  Diffusion-KTO  | 21.14 | 56.03 |33.81 |27.31 | 52.22
> |  Self-DPO   |21.23 | 56.35 | 34.79 | 27.33 | 71.00
>
> Our method demonstrates superior performance over Diffusion-KTO. Notably, while Diffusion-KTO avoids pairwise ranking, it still requires binary annotations (e.g., "like/dislike") for training, which incurs annotation costs similar to traditional preference-based methods.
> Thank you for your valuable feedback.
>
> **W2 & Q2:More ablation studies on fine-tuning methods used and its interaction effect with MSE losses are needed to clarify the effectiveness of each component; The paper does not clearly specify the fine-tuning method used.**
>
> In the main text, all experiments are conducted under full-parameter fine-tuning. Here, we present results using LoRA fine-tuning with rank 64:
>
>
> |   Models    | PickScore | Aesthetics | CLIP | HPS | ImageReward
> |-----------------------|-----------|-----------|-----------|-----------|-----------|
> |  SD1.5 | 20.57 |53.15|32.58 |26.17| -14.81
> |  SD1.5 + DPO (Full)   |20.91 |54.07| 33.19|26.46 |4.13
> |  SD1.5 + Self-DPO (Full)  |21.23 |56.35| 34.79 |27.33 |71.00
> |  SD1.5 + DPO (LoRA)   |20.62| 53.34 |32.60 |26.28 | -11.63
> |  SD1.5 + Self-DPO (LoRA)  |20.95| 54.77 |34.38 |26.85 | 23.82
>
> We observe that both DPO and Self-DPO under LoRA show lower performance than their full-parameter counterparts but outperform the baseline.
> This can be attributed to the fact that LoRA reduces trainable parameters significantly compared to full fine-tuning. Notably, Self-DPO consistently surpasses DPO in all metrics, demonstrating its robustness even with parameter constraints.
>
> **Q3:This paper's method is simple and clean, which is good. The effectiveness of the proposed method needs more evidence.**
>
> We further validate our method’s effectiveness by introducing a new real-world T2I evaluation setting. Specifically, we conduct experiments on a subset of the LAION-COCO dataset containing ~400k text-image pairs. Unlike carefully curated datasets, we do not manually design the data distribution or employ re-captioning techniques to refine prompts.
>
> |   Models    | PickScore | Aesthetics | CLIP | HPS | ImageReward
> |-----------------------|-----------|-----------|-----------|-----------|-----------|
> |  SD1.5 | 20.57 |53.15|32.58 |26.17| -14.81
> |  SD1.5 + SFT   |20.49| 52.63 |31.83 |26.13| -23.95
> |  SD1.5 + Self-DPO  |20.68| 52.87 |34.31 |26.52 | 17.11
>
>
> While naive SFT degrades performance, our Self-DPO method achieves consistent improvements over the baseline across four out of five metrics.
> Please note that we cannot provide a DPO baseline as it requires additional paired images and annotations for the same prompts, which are not available in regular text-image dataset.
>
> **Formatting Concerns: Rows 168 and 169 are too close.**
>
> We will fix it in the revision. Thank you for your advice.

---

> > ### Comment · Reviewer_vbGM · 2025-08-03
> >
> > I appreciate the author's efforts in the rebuttal. I will increase the rating accordingly.

---

### Official Review · Reviewer_viTx · 2025-07-02

**Clarity:** 3
**Significance:** 2
**Originality:** 2
**Rating:** 4
**Confidence:** 4

**Summary:**

The authors propose Self-DPO, a direct alignment algorithm that curates image preference data through image augmentation. Self-DPO addresses the reliance on paired preference data required by Diffusion DPO, enabling direct alignment with high-quality images and their captions. The authors validate the effectiveness of Self-DPO on Stable Diffusion v1.5 and Stable Diffusion XL, demonstrating superior performance compared to Diffusion DPO and SFT across several automatic evaluation metrics.

**Questions:**

* It seems that the MSE term is doing SFT and did not impact the performance significantly, it would be interesting to see how tuning the weights of MSE and Self-DPO affects the performance.
* On SDXL, given that the MSE term is essentially doing SFT and could degrade performance, has the authors attempted to remove the MSE term and keep only the Self-DPO loss?
* The scale of reported Image Reward seems different from the raw model output, did the authors scale it by 100?
* Could the authors potentially include VLM evaluation on general human preference to complement the reward models?

**Ethical Concerns:**

["NO or VERY MINOR ethics concerns only"]

**Final Justification:**

As suggested in my comments

**Limitations:**

yes

**Paper Formatting Concerns:**

As discussed in Questions

**Quality:**

3

**Strengths And Weaknesses:**

* Strengths:
  * Simple solution to an existing data dependence problem to Diffusion-DPO, easy to implement for practitioners.
  *  Comprehensive evaluation on automatic models with superior or at least comparable results to DPO without paired data.
  * Writing is clear and easy to follow.
* Weakness:
  * The proposed algorithm seems simply borrows existing negative sample crafting techniques from SSL, the technical contribution is limited.
 * The experiments are limited to SD1.5 and SDXL, limiting the robustness of Self-DPO.
 * The authors only evaluated CLIP-based reward model suites, additional human/ large VLM evaluation could further solidate the results.

---

> ### Author Rebuttal · Authors · 2025-07-30
>
> **W1: The proposed algorithm seems simply borrows existing negative sample crafting techniques from SSL, the technical contribution is limited.**
>
> We appreciate your feedback and acknowledge that our implementation adopts a straightforward approach. However, we emphasize two key distinctions from SSL. First, our work introduces a novel perspective to direct preference learning by exploring whether preference data pairs can be generated directly from unpaired data. We affirmatively answer this question through a simple yet effective framework, demonstrating that this approach not only achieves good performance but also significantly reduces the annotation cost inherent in traditional DPO methods. Second, task-relevant image degradation plays a critical role, as demonstrated by our ablation study in Table 3 of the main text. Different degradation strategies yield divergent performance outcomes, and we propose a simple yet effective degradation method. Crucially, we show that excessive image quality degradation negatively impacts results, underscoring the need for balanced degradation design.
>
> **W2: The experiments are limited to SD1.5 and SDXL, limiting the robustness of Self-DPO.**
>
> Prior works commonly adopt two widely-used baselines (SD1.5 and SDXL), and we follow this fashion. Our method is model-agnosic and can be employed in most diffusion models.  As an initial attempt, we extend our evaluation to the state-of-the-art FLUX model using the Pick-a-Pic V2 synthetic dataset. Notably, this dataset comprises image generations from SD2.1 and SDXL, which have lower quality compared to FLUX outputs. Despite this, our experiments demonstrate that the method remains effective when applied to modern architectures.
>
> |                 | PickScore | Aesthetics | CLIP | HPS | ImageReward
> |-----------------------|-----------|-----------|-----------|-----------|-----------|
> | Flux    | 22.45     | 62.25 | 32.95 | 28.81 | 102.00
> | Flux + DPO  | 22.46      |  **62.38** | 32.88 | 28.86 | 101.41
> | Flux + Self-DPO  | **22.52**     |  61.98 | **33.56** | **28.99** | **108.31**
>
> **W3 & Q4: The authors only evaluated CLIP-based reward model suites, additional human/ large VLM evaluation could further solidate the results; Could the authors potentially include VLM evaluation on general human preference to complement the reward models?**
>
> We employ UnifiedReward [1] as the VLM evaluation metric, a state-of-the-art reward model designed for multimodal understanding and generation tasks. Built upon Qwen2.5-VL-Instruct, it supports pointwise scoring to align model outputs with human preferences. The experimental results are summarized below:
>
> SD1.5 resluts:
> |                 | SD1.5 | SFT | DPO | Self-DPO
> |-----------------------|-----------|-----------|-----------|-----------|
> | UnifiedReward  | 2.44    | 2.62 | 2.53 | **2.72**
>
> SDXL resluts:
> |                 | SDXL | SFT | DPO | Self-DPO
> |-----------------------|-----------|-----------|-----------|-----------|
> | UnifiedReward  | 2.96   | 2.73 | **3.10** | 3.07
>
> FLUX resluts:
> |                 | FLUX  | DPO | Self-DPO
> |-----------------------|----------|-----------|-----------|
> | UnifiedReward  | 3.20 | 3.23  | **3.28**
>
>
> The results demonstrate that Self-DPO shows better results over baseline methods under most comparisons. It highlights the effectiveness of our proposed method in aligning model outputs with human preferences.
>
> [1] Unified Reward Model for Multimodal Understanding and Generation. arXiv 2025.
>
> **Q1: It seems that the MSE term is doing SFT and did not impact the performance significantly, it would be interesting to see how tuning the weights of MSE and Self-DPO affects the performance.**
>
> We present an initial ablation study on the impact of removing the MSE loss in the main text. Here, we provide an expanded analysis comparing different weight ratios between the Self-DPO and MSE components:
>
> |      Self-DPO : MSE         | PickScore | Aesthetics | CLIP | HPS | ImageReward
> |-----------------------|-----------|-----------|-----------|-----------|-----------|
> |  0.00 : 1.00   | 21.10 | 56.35 |33.75 |27.03 |45.03
> |  0.25 : 0.75   |21.19| 56.23 |34.71 |27.29 |66.20
> | 0.50 : 0.50  | 21.23 |56.35 |34.79 |27.33 |71.00
> | 0.75 : 0.25  | 21.24 | 56.24 |34.75 |27.34| 68.27
> | 1.00: 0.00 |21.26| 56.22 |35.00 |27.31 |67.78
>
> We have two obervations.
> First, when Self-DPO is entirely removed (0.0:1.0), the method reduces to standard SFT, yielding the lowest performance across most metrics.
> Second, introducing Self-DPO (ratios > 0.0:1.0) consistently improves results, with performance remaining stable across varying weight distributions. This suggests the Self-DPO loss plays the main role during the training process.
> These findings confirm that our method is insensitive to loss weight tuning, as the Self-DPO component primarily delievers performance gains.
>
> **Q2: On SDXL, given that the MSE term is essentially doing SFT and could degrade performance, has the authors attempted to remove the MSE term and keep only the Self-DPO loss?**
>
> Thanks for your advice! We conducted this experiment, and the results are presented below:
>
> |                 | PickScore | Aesthetics | CLIP | HPS | ImageReward
> |-----------------------|-----------|-----------|-----------|-----------|-----------|
> | SDXL    | 22.10 | 60.01 | 35.86 | 26.83 | 50.62
> | SDXL + DPO  | 22.57 | 59.93 | 37.30 | 27.30 | 81.14
> | SDXL + Self-DPO  | 22.34 | 59.97 | 37.53 | 27.89 | 103.96
> | SDXL + Self-DPO (No MSE) | 22.42     |  59.78 | 37.52 | 28.07 | 106.37
>
> As anticipated, eliminating the MSE loss for the SDXL baseline yields marginal improvements. This aligns with our hypothesis that the Self-DPO loss plays the main role, and its standalone application achieves the best overall performance. Thank you for your valuable feedback!
>
> **Q3: The scale of reported Image Reward seems different from the raw model output, did the authors scale it by 100?**
>
> Yes. As shown in Line 203-204 in the main text, we scale ImageReward by a factor of 100. It is beacause we aim to adjust the value range approximately 0-100.

---

> > ### Author Response · Authors · 2025-08-04
> > **A gentle reminder**
> >
> > Dear Reviewer viTx,
> >
> > As the NeurIPS author-reviewer discussion period approaches its conclusion, we would like to kindly follow up on our rebuttal. We sincerely appreciate your time and effort in reviewing our work. We would be grateful if you could confirm whether our response adequately addressed your concerns. If further clarifications are needed, please let us know. We are happy to provide additional details. Should your concerns have been resolved, we would greatly appreciate it if you could consider revising your review score accordingly. Thank you again!

---

### Decision · Program_Chairs · 2025-09-17

**Decision:**

Accept (poster)

**Comment:**

This paper proposes a refinement/simplification to the Diffusion DPO algorithm that solves the data dependency issue. Rather than relying on human-labeled preference pairs, they generate synthetic pairs by taking high quality images and downgrading them in order to constructing winning/losing pairs. They show that this exceeds the effectiveness of regular DPO.

Reviewers had no major objections, they found the approach elegant. There was some concerns that certain ablations were not done, but the authors seemed to respond effectively. I personally have 2 concerns that I think I take more seriously than the reviewers do:

a. Evaluation was done by (weak) reward models not stronger VQA and Human ratings, and these frequently do not transfer to downstream effectiveness.
b. I don't think this sort of augmentation can completely replace the usual way of doing preference set construction for DPO, we do need to sample a fuller distribution of deltas between winner and loser, there is a chance in fact that "simpler" degradations are exactly the ones that will give gains on CLIP without reflecting in gains or harder evaluations.

However, I think since this paper was universally approved by the reviewers, I still recommend acceptance.